# Comparison of the Effects of Propofol–Dexmedetomidine and Thiopental–Dexmedetomidine Combinations on the Success of Classical Laryngeal Mask Airway Insertions, Hemodynamic Responses, and Pharyngolaryngeal Morbidity [note 1]

**DOI:** 10.3390/medicina61050783

**Published:** 2025-04-23

**Authors:** Mensure Çakırgöz, İsmail Demirel, Aysun Afife Kar, Ergin Alaygut, Ömürhan Saraç, Emre Karagöz, Oğuzhan Demirel, Mert Akan

**Affiliations:** 1Intensive Care Unit, Department of Anesthesiology and Reanimation, Izmir Tepecik Training and Research Hospital, University of Health Sciences, Izmir 35330, Turkey; 2Intensive Care Unit, Department of Anesthesiology and Reanimation, Firat University School of Medicine Hospital, Elazig 23119, Turkey; 3Intensive Care Unit, Department of Anesthesiology and Reanimation, Izmir City Hospital, University of Health Sciences, Izmir 35540, Turkey; aaysunkar@hotmail.com (A.A.K.);; 4Department of Anesthesiology and Reanimation, Ankara University Ibni Sina Hospital, Ankara 06230, Turkey; 5Intensive Care Unit, Department of Anesthesiology and Reanimation, Izmir Acibadem Kent Hospital, Izmir 35620, Turkey

**Keywords:** dexmedetomidine, propofol, thiopental, laryngeal mask airway

## Abstract

*Background and Objectives*: Dexmedetomidine is a potent selective α_2_ receptor agonist with analgesic and sedative effects. Many reports indicate that compared to fentanyl, the combination of dexmedetomidine with propofol provides comparably acceptable conditions for a laryngeal mask airway (LMA). However, no study has evaluated the effectiveness of combined dexmedetomidine and thiopental in LMA insertions compared to that of combined dexmedetomidine and propofol. This prospective, randomized, double-blind study aimed to compare the effects of dexmedetomidine with thiopental or propofol on LMA insertion conditions, hemodynamic responses, and pharyngolaryngeal morbidity, which in this study was defined as the presence of postoperative sore throat, dysphagia, or visible blood in the airway following a laryngeal mask airway (LMA) insertion. *Materials and Methods*: A total of 80 premedicated ASA I-II patients aged 18–65 years were randomized to the propofol group (Group P, *n* = 40) or thiopental group (Group T, *n* = 40). Anesthesia was induced by infusing 1 μg·kg^−1^ dexmedetomidine over 10 min followed by 2.5 mg·kg^−1^ propofol or 5 mg·kg^−1^ thiopental. LMA insertion conditions were evaluated on a scale assessing six variables. Systolic arterial pressure (SAP), diastolic arterial pressure (DAP), mean arterial pressure (MAP), heart rate (HR), and bispectral index values were recorded at baseline; 1 min before; and at 1, 2, 3, 4, and 5 min after an LMA insertion. The baseline values for the systolic arterial pressure (SAP), diastolic arterial pressure (DAP), mean arterial pressure (MAP), heart rate (HR), and bispectral index (BIS) values were recorded before dexmedetomidine infusion. Measurements for all patients were then taken 1 min before and at 1, 2, 3, 4, and 5 min after the LMA insertion *Results*: Demographic data were similar between the groups. In Group P, the time to loss of eyelash reflex and LMA insertion time were significantly shorter, the apnea duration was significantly longer, and the rates of full jaw opening and optimal LMA insertion conditions were significantly higher when compared with those of Group T (*p* < 0.05). Group P showed a significantly greater percentage decrease in HR compared to that of Group T at 1 min before and 1, 2, and 3 min after the LMA insertion (*p* < 0.05). Group T had a greater decrease in SAP and MAP at 1 min before insertion, while the SAP decrease was lower in Group T at 3, 4, and 5 min after insertion. The MAP and DAP values after the LMA insertion showed a greater decrease in Group P compared to in Group T (*p* < 0.05). The incidence of bradycardia was significantly (*p* < 0.05) higher in Group P than in Group T. There was no significant difference between the groups in terms of the frequency of hypotension, sore throat, presence of blood, or dysphagia at discharge from the recovery unit (*p* > 0.05). *Conclusions*: This study showed that the use of dexmedetomidine with thiopental provided comparably acceptable LMA insertion conditions with more stable hemodynamics compared to propofol.

## 1. Introduction

The role of the anesthesiologist includes many responsibilities, one of which is ensuring a patient’s airway is maintained during surgical procedures [1]. The laryngeal mask airway (LMA) is one of the most frequently used noninvasive airway devices in anesthesia practice, especially for outpatient surgeries. It is blindly placed in the hypopharynx, enabling supraglottic delivery of oxygen or anesthetic gases with spontaneous or positive pressure ventilation [2,3].

Proper insertion of the LMA is important for the airway and patient safety because reflex responses such as swallowing, coughing, or gagging can interfere with the placement of the LMA or cause a correctly positioned LMA to dislodge [4]. A successful LMA insertion depends on achieving sufficient mouth opening and an adequate depth of anesthesia to suppress reflexes such as coughing, gagging, and laryngospasm [5]. To this end, many studies have been conducted to determine the agent and induction techniques that provide optimal conditions. Propofol has become the most frequently preferred induction agent because it suppresses pharyngeal and laryngeal responses better than thiopental and other induction agents [1,5,6,7,8,9]. However, when used alone in patients without premedication, it requires a bolus dose (2.5–3.0 mg·kg^−1^) or high target plasma concentration (7–9 μg·mL^−1^) that can result in severe cardiovascular depression and prolonged apnea [10]. As a result, the use of propofol alone at the recommended induction doses does not guarantee a successful LMA insertion [5,6,7,10,11]. Thiopental is less costly and provides similar LMA insertion conditions to propofol without an increase in apnea duration or cardiovascular depression. However, as it cannot provide the sufficient conditions for a successful LMA insertion when used alone, it has been used with numerous adjuvants [1,5,6,7,12]. Therefore, it is the most frequently used agent in studies utilizing the coinduction method [1,6,12].

Dexmedetomidine, a pharmacologically active dextro isomer of medetomidine, is a highly selective, potent, and specific α_2_-adrenoceptor agonist with an elimination half-life of 2.1–3.1 h [1,5,6,7,13,14,15,16]. Many reports have indicated that the administration of dexmedetomidine before induction with propofol results in excellent or acceptable LMA insertion conditions at rates of 90–100% [2,13,17,18,19]. However, we encountered no study in the literature evaluating the effect of administering dexmedetomidine with thiopental, which has a lower risk of hemodynamic depression than propofol, on LMA insertion conditions. Therefore, in this study, we aimed to compare the effect of 1 μg·kg^−1^ dexmedetomidine administered before induction with 2.5 mg·kg^−1^ propofol or 5 mg·kg^−1^ thiopental on LMA insertion conditions, as well as pharyngolaryngeal morbidity criteria, including dysphagia, blood on the LMA, and sore throat, in patients undergoing surgery under general anesthesia lasting less than 2 h and not requiring intubation.

## 2. Materials and Methods

This study received permission from the the Okmeydani Education and Research Hospital Medication Research Local Ethics Committee (Istanbul, Turkey) (date 3 September 2013, decision number 109). This study was completed in the surgeries of the Firat University Medical Faculty and Okmeydani Education and Research Hospital after receiving informed consent from patients between 1 October 2013 and 1 April 2014. A total of 80 patients who were between the ages of 18 and 65 years, were in class I-II according to the American Society of Anesthesiologists (ASA) physical status classification system, were undergoing elective surgery not exceeding 2 h in duration, did not require a muscle relaxant, and had an indication for an LMA insertion were included in this study. Patients with any of the following were excluded: neck or upper respiratory tract pathology, sore throat, dysphagia, or dysphonia; history or risk factors for a difficult airway (Mallampati class 3–4, sternomental distance less than 12 cm, thyromental distance less than 6 cm, head extension less than 90 degrees, mouth opening less than 1.5 cm); morbid obesity; or a history of lung disease, alcohol or substance addiction, chronic sedative or opioid analgesic use, or an allergy to the study drugs.

All patients underwent standard monitorization consisting of heart rate (HR), systolic arterial pressure (SAP), diastolic arterial pressure (DAP), mean arterial pressure (MAP), electrocardiography (ECG-derivation II), and peripheral oxygen saturation (SpO_2_) in the operating room before anesthesia induction. The depth of anesthesia was checked by monitoring the bispecteral index (bis-Vista^TM^; Aspect Medical Systems, Newton, MA, USA) [20]. A 20-gauge intravenous (IV) cannula was inserted via a dorsal vein of the hand, and a 7 mL·kg^−1^ saline infusion was administered before induction [21]. The study drugs were administered to all patients following preoxygenation for 3 min [2,4]. Using the sealed envelope method, the patients were randomly assigned to receive 1 μg·kg^−1^ dexmedetomidine followed by 2.5 mg·kg^−1^ propofol (Group P, *n* = 40) or 5 mg·kg^−1^ thiopental (Group T, *n* = 40). The induction sequence was performed using two syringes prepared in advance. The first syringe contained 1 µg·kg^–1^ dexmedetomidine and was filled with saline to a total volume of 50 mL. The second syringe contained 20 mL of 1% (10 mg/mL) propofol or 2% (20 mg/mL) thiopental and was covered with opaque tape to obscure its contents. To ensure a double-blind study protocol, the coded test syringes were prepared by a nurse who was not involved in this study. In addition, all syringes were injected by a resident behind a drape so that the anesthesiologist inserting the LMA and monitoring the study parameters remained blind to the drug doses [21,22]. Thus, drug preparation and administration and LMA insertion and follow-up were conducted by different people. This ensured that the anesthesiologist who inserted the LMA, evaluated the patient’s responses, and monitored/recorded the study parameters was blinded to the drugs given [21]. The 1 µg·kg^−1^ dexmedetomidine was infused over 10 min using an infusion pump (Braun Infusomat ^®^; Braun Melsungen Ko, Melsungen, Germany) [4,17,18,23,24]. Three minutes before induction, premedication with 0.03 mg·kg^−1^ IV midazolam (Dormicum^®^ ampoule, Roche preparations Industry, Istanbul, Turkey) was administered [6]. In the last 30 s of the dexmedetomidine infusion, a hypnotic agent at a fixed dose of 0.25 mL·kg^−1^ was infused over 30 s for anesthesia induction [4,12,17,22,25].

After induction, the patients were ventilated with pure oxygen via a face mask and their eyelash reflex was checked [26]. The duration from the start of the induction agent to the loss of the eyelash reflex (loss of eyelash reflex duration: LED) was recorded [8,20]. At 90 s after induction [2,11,12,17,18,21,25,26], when the BIS value was less than 40 [20] and sufficient jaw relaxation was achieved, the surface of the LMA facing the oropharynx was lubricated using a water-soluble gel and the LMA was inserted with the cuff completely deflated as per the standard method described by Brain. The same researcher with more than 3 years of experience performed the LMA insertion in all patients [6,16,18,19,22]. After the placement of the LMA, cuff pressure was monitored (cuff pressure manometer; Rüsch, Germany) to standardize postoperative pharyngeal morbidity. The cuff was inflated with air to a pressure of 60 cmH_2_O, which was kept constant throughout the operation [27,28].

The successful LMA insertion duration (ID: measured from the mouth opening to the first successful ventilation) was recorded [16]. A successful LMA placement was defined based on the following criteria: a square wave pattern observed on the capnogram, comfortable bag–valve–mask ventilation, visible chest movements, and no air leakage during ventilation with a maximum positive pressure of 20 cmH_2_O [2,4,16,17,18,29]. Anesthesia was maintained with 1 Minimum Alveolar Concentration (MAC) sevoflurane in 40% O_2_/60% N_2_O mixture [12,17,18,22,24]. The sevoflurane concentration was adjusted to maintain a BIS value between 40 and 60 [20,30]. If adequate induction could not be achieved, if any movement was observed during the first attempt, or if necessary to maintain a BIS value below 40, an additional dose corresponding to 0.125 mL/kg of each study drug was administered and an LMA insertion was attempted a second time 60 s later [11,20,22]. The number of attempts was recorded, but the conditions for the LMA insertion were evaluated only during the first attempt [11,16]. If insertion was unsuccessful in two attempts (because of inadequate ventilation or air leakage that cannot be resolve by repositioning, hypercarbia [end-tidal CO_2_ > 45), or hypoxemia (SpO_2_ < 90% or complete laryngospasm at any time during this study]), intubation was performed and the patient was excluded from this study [4,11,19,20,22,29]. The conditions for the LMA insertion were evaluated using a 6-variable scale (jaw opening, ease of insertion, swallowing, coughing/gagging, laryngospasm, and movement) as in previous studies. The LMA insertion conditions were evaluated as “optimal” if all the criteria were excellent, “acceptable” if there was a mix of excellent and good criteria, and “poor” if one or more of the criteria were poor [31] (Table 1).

The baseline values for SAP, DAP, MAP, HR, BIS, and SpO_2_ were recorded immediately after preoxygenation and prior to the initiation of the dexmedetomidine infusion. Following this, measurements were taken for all patients at 1 min before and at 1, 2, 3, 4, and 5 min after the LMA insertion. The changes through time for HR, MAP, and SAP were assessed by calculating the percentage change from the baseline. The apnea duration (AD) was determined (defined as the time from the last spontaneous breath after induction to the first spontaneous breath) [2,4,12,17,20]. Then, %100 oxygen was initiated 5 min before the end of surgery. Before removing the LMA, the intra-cuff pressure was measured and recorded. When ventilation was adequate, the LMA was removed and the duration of the LMA use (use duration: UD) was recorded as the time from insertion to removal [28]. After removing the LMA, the presence of blood was assessed on a scale of 1–3 as none, trace, and significant, respectively [32]. The patient was awakened and transferred to the recovery unit with pure oxygen. To determine the frequency and severity of pharyngolaryngeal complications, all patients were evaluated in the recovery unit for sore throat (constant pain, independent of swallowing) and dysphagia (difficulty swallowing provoked by drinking) by a single researcher who was blind to their group allocation and not involved in the anesthesia process. Patients were specifically asked about the presence/absence of these symptoms postoperatively. Sore throat was rated on a scale of 0–3 as none, mild, moderate, and severe, respectively [28,33]. Peroperative hypotension (<30% decrease in MAP compared to baseline value) was treated by administering 6 mg of ephedrine (Ephedrine, Haver, Istanbul, Turkey). Bradycardia was defined as HR below 50 beats/min and was treated with 0.5 mg IV atropine (Atropine Sulfate, Haver, Istanbul, Turkey) [21].

**Statistical methods:** Descriptive statistics (mean, standard deviation, median, minimum, maximum) were used to describe continuous variables. The distribution of categorical variables was presented using frequencies and percentages. The normality of the data distribution was assessed using the Shapiro–Wilks test. Comparing two independent continuous variables that did not follow a normal distribution was analyzed using the Mann–Whitney U test. A paired sample *t* test was used in the comparison between two dependent continuous variables. The relationship between the categorical variables was examined using the chi-square test (or Pearson/Fisher’s exact test, where appropriate). The effect of time in groups was examined with a repeated measures ANOVA test. A *p*-value of less than 0.05 was considered statistically significant. The analyses were performed using MedCalc Statistical Software version 12.7.7 (MedCalc Software bvba, Ostend, Belgium; http://www.medcalc.org; 2013).

## 3. Results

The two groups had similar demographic characteristics (Table 2). Patients included in this study underwent operations lasting less than 2 h (averagely 1 h) from general surgery, gynecology, obstetrics, or urology. The time to eyelash reflex loss and LMA placement time were significantly lower in Group P compared to in Group T (*p* < 0.05). The apnea duration was significantly longer in Group P compared to in Group T (*p* < 0.05) (Table 3). The rates of full jaw opening and optimal LMA insertion conditions were significantly higher in Group P compared to in Group T (*p* < 0.05). There were no significant differences between the groups in terms of the operative time, duration of LMA use, ease of LMA insertion, acceptable LMA insertion conditions, movement, swallowing, coughing/gagging, laryngospasm, or having two or more insertion attempts (Table 4).

There was no significant difference in the baseline HR, SAP, DAP, or MAP values between the groups (*p* > 0.05). Compared to the baseline values, HR, SAP, DAP, and MAP were significantly lower in both groups during this study (*p* < 0.05). Group P showed a significantly greater percentage decrease in HR compared to Group T at 1 min before and 1, 2, and 3 min after LMA insertion (*p* < 0.05). Group T showed a significantly greater percentage decrease in SAP and MAP compared to Group P at 1 min before insertion. Group T showed a significantly lower percentage decrease in SAP compared to Group P at 3, 4, 5 min after insertion. The MAP and DAP values at all time points after LMA insertion showed a significantly greater percentage decrease in Group P compared to in Group T (*p* < 0.05) (Table 5). The incidence of bradycardia was significantly (*p* <0.05) higher in Group P than in Group T. There was no significant difference between the groups in terms of the frequency of hypotension and postoperative sore throat, presence of blood, or dysphagia (*p* > 0.05) (Table 6).

## 4. Discussion

Based on our review of the literature, our study is the first to evaluate the effect of the preinduction administration of dexmedetomidine with thiopental on LMA insertion conditions and to compare this to the combination of dexmedetomidine and propofol. The results of our study showed that in healthy, premedicated patients with appropriate airway anatomy, 1 μg·kg^−1^ dexmedetomidine followed by thiopental provided optimal or acceptable LMA placement conditions at similar rates to the dexmedetomidine/propofol combination (95% vs. 100%). In addition, although the differences were not clinically significant, thiopental was associated with the longer time to eyelash reflex loss and LMA placement time, the shorter apnea duration, the more stable hemodynamics without an increased risk of bradycardia, and a lower rate of optimal LMA insertion conditions (40% vs. 62.5%) compared to propofol.

Various methods have been used to determine the equipotential doses of anesthetics. Although there is controversy about the exact equivalents, many comparative studies used a 2:1 ratio and demonstrated that 2.5 mg·kg^−1^ propofol and 5 mg·kg^−1^ thiopental are equipotent doses [5,6,7,8,9,25]. Similarly, in a study examining the electroencephalogram (EEG) changes associated with a loss of consciousness during induction, the same dose ratio was used for propofol and thiopental [21]. However, another study comparing the effects of induction agents on the cardiac index reported that when loss of the lash reflex was used as the induction endpoint, the average doses of induction agents were 2.24 mg·kg^−1^ for propofol and 4.75 mg·kg^−1^ for thiopental [34]. Therefore, for induction in our study we opted to use the doses of 5 mg·kg^−1^ thiopental and 2.5 mg·kg^−1^ propofol used in previous comparative studies.

Preadministration of dexmedetomidine causes a dose-dependent reduction in the anesthetic requirement for successful insertion of supraglottic airway devices, although the rate or timing of administration may affect the amount of anesthetic needed [4,14,35]. A rapid injection of dexmedetomidine can shorten the onset of action, leading to a further reduction in the need for anesthetics and an increase in the risk of hypertension and/or bradycardia, while continuous infusion may delay recovery due to its sedative properties [4,19,36]. In our study, we preferred a single dose of 1 μg·kg^−1^ dexmedetomidine delivered as a 10 min infusion before induction based on previous reports that this dose protocol effectively suppressed the hemodynamic response to intubation and/or LMA insertion, causing less respiratory depression and prolonged stabilization of heart rate and blood pressure slightly below the baseline [4,17,18,23,24,36].

When propofol and thiopental are administered alone at the recommended induction doses, insufficient suppression of airway reflexes such as swallowing, gagging, coughing, and laryngospasm and movement responses inevitably lead to undesirable events during an LMA insertion [5,7,10]. For this reason, coinduction techniques in which various agents are used as adjuvants to improve LMA insertion conditions and reduce the necessary dose have been applied [1,5,6,7,8,17,20,31,36]. Coinduction is the practice of combining different anesthetic agents in smaller doses in order to achieve similar induction conditions to those obtained when induction agents are used alone at high doses while avoiding the adverse effects associated with those high doses [37]. However, to achieve the best possible insertion conditions when practicing coinduction, placement of the LMA should coincide with the peak effects of both the anesthetic and adjuvant agents. Thiopental, propofol, and midazolam reach peak plasma and effect-site concentration approximately 1.6, 1.75, and 4 min after IV administration, respectively [6,12]. For this reason, we administered midazolam 3 min before propofol and thiopental administration and inserted the LMA 90 s after that [6,21].

Dexmedetomidine is a highly selective and potent α2-adrenoceptor agonist that has dose-dependent sedative, anxiolytic, analgesic, and sympatholytic effects and is frequently preferred for outpatient surgeries due to its short duration of action [2,13,14,17,18,19,23]. In a study evaluating the effect of 5 mg·kg^−1^ thiopental and 2.5 mg·kg^−1^ propofol on airway reactivity, McKeating et al. [8] reported that compared to thiopental, the propofol group had lower airway reactivity (0% vs. 27%) and higher rates of jaw relaxation (100% vs. 84%) and successful laryngoscopy (100% vs. 66%). Bolcuoğlu et al. [25] compared intubation conditions without a muscle relaxant after administering dexmedetomidine followed by induction with propofol, thiopental, or etomidate. They reported that none of the patients in the etomidate group could be intubated and the highest rate of intubation success was achieved with propofol, although there was no statistical difference from thiopental (53.3% vs. 30%). However, many studies have shown that the coadministration of dexmedetomidine and propofol increases the success of a fiberoptic intubation, awake blind nasotracheal intubation, LMA insertion, and tracheal intubation using a muscle relaxant, with an increased risk of bradycardia [2,18,19,25,30,38,39]. However, we found no study evaluating LMA insertion conditions with the combination of dexmedetomidine and thiopental, which causes less hemodynamic depression than propofol. In our study, we found that Group T had a statistically significantly lower rate of full mouth opening (55% vs. 77.5%) and excellent laryngeal mask placement conditions (40% vs. 62.5%) compared to Group P (*p* < 0.05). However, both groups achieved similar rates of excellent or satisfactory laryngeal mask placement conditions (95% vs. 100%). Laryngospasm occurred in two patients in the thiopental group but none in the propofol group. The results of our study showed that when administered with dexmedetomidine, both agents resulted in a comparable level of optimal or acceptable LMA insertion conditions with an increase in LMA insertion success due to the suppression of pharyngolaryngeal reflexes. However, consistent with the literature data, as thiopental causes less depression of muscle tone and upper airway reflexes, its difference from propofol was reflected in terms of complete jaw relaxation and optimal LMA insertion conditions [1,5,6].

McCollum and Dundee [40] reported that in premedicated, untreated patients, the administration of 2.0 or 2.5 mg·kg^−1^ propofol caused greater reductions in SAP after induction than 5 mg·kg^−1^ thiopental (15%, 17%, and 10%, respectively) and caused hypotension. Dexmedetomidine causes a dose-dependent decrease in blood pressure after IV bolus administration, but previous reports have reported that pretreatment with dexmedetomidine reduces the decrease in blood pressure during propofol induction both before and after LMA placement [4,14,17,18,35]. A meta-analysis study by Casaai et al. [38] evaluated the effect of dexmedetomidine on the hemodynamic response to tracheal intubation. The authors reported that postinduction SAP and MAP were lower with thiopental compared to propofol, showing that the selected induction agent affected the hemodynamic profile of dexmedetomidine. Consistent with the results of previous research, pretreatment with dexmedetomidine attenuated the decrease in blood pressure seen during propofol induction in our study. When compared with the baseline values, the relative reduction in SAP and MAP after induction was 17% and 14.6% in the thiopental group compared to 10.2% and 10.3% in the propofol group, respectively. Following LMA insertion, an increase in blood pressure over the baseline was not observed in either group, while the HR, SAP, DAP, and MAP values continued to decrease.

Dexmedetomidine typically produces a biphasic and dose-dependent hemodynamic response that causes hypotension at low plasma concentrations and hypertension at high plasma concentrations. Rapid and/or high-dose dexmedetomidine administration results in a hypertensive phase in which hypertension is accompanied by bradycardia due to peripheral vasoconstriction resulting from the activation of α_2_ receptors in the vascular smooth muscle and the baroceptor reflex. After a few minutes, the vasoconstriction abates due to a reduction in the plasma concentration, while the simultaneous activation of α_2_ adrenoreceptors in the vascular endothelial cells causes vasodilation. However, the postsynaptic activation of α_2_ adrenoceptors in the central nervous system results in a hypotensive phase involving hypotension due to increased cardiovagal activity with decreased sympathetic activity, accompanied by bradycardia [14,35]. Since the hypertensive response is often associated with rapid injection, we believe that we did not observe the expected biphasic effect with a hypertensive response because we administered dexmedetomidine as an infusion over 10 min [30].

Dexmedetomidine has been shown to suppress the hemodynamic response to stimuli such as laryngoscopy, intubation, and LMA insertions with enhanced hemodynamic stability when administered perioperatively [14,17,30]. The most common side effects are hypotension and bradycardia, which are mostly seen during the loading period due to its sympatholytic and vagotonic effects [2,4,18,24,30,36,39]. The cardiovascular depressant effect of propofol is a direct result of myocardial depression and decreased systemic vascular resistance, and it is also characterized by a lower rise in HR despite a significant decrease in arterial pressure associated with suppression of the baroreflex mechanism [26,34]. These negative chronotropic effects are the result of a transient increase in the parasympathetic tone relative to the sympathetic tone, thus explaining the increased risk of bradycardia and/or hypotension with coadministration of dexmedetomidine and propofol [2,18,26,31,36]. In a study by Choudhary et al. [18] that used a similar induction protocol to ours and evaluated the effect of fentanyl or dexmedetomidine before induction with propofol on the success of LMA insertion, bradycardia was observed in a total of seven patients (one of whom had both bradycardia and hypotension), yielding rates similar to those of our study (17.5% and 13.5%, respectively). However, in similar studies premedicating with an anticholinergic agent such as glycopyrrolate or atropine, the incidence of hypotension and/or bradycardia has been reported to be 0–4% [13,17,23]. For this reason, we believe that the high incidence of bradycardia in our propofol group consistent with previous studies may be attributable to not premedicating with an anticholinergic agent. However, in a study by Liu et al. [36] examining the effects of different doses of dexmedetomidine (0.25, 0.5, or 1 μg·kg^−1^) infused over 10 min before propofol induction on hemodynamic responses to laryngoscopy, anesthesia requirement, and recovery, bradycardia was reported in 5 of 20 patients (a rate of 25%) in the group treated with 1 μg·kg^−1^ dexmedetomidine. They suggested this may be related to the increased vagal tone and therefore recommended a dose of 0.5 μg·kg^−1^ dexmedetomidine to reduce the risk of bradycardia. In our study, we found that the incidence of bradycardia was significantly higher with propofol than thiopental. In the thiopental group, bradycardia was not observed in any patient, consistent with the results of Gögüs et al. [24]. Based on previous reports and the current findings, clinicians should consider the risk of bradycardia when combined with propofol.

The sedative and hypnotic effect of dexmedetomidine is dose-dependent and resembles natural sleep, with minimal effect on respiration and ventilation, and it occurs through inhibition of norepinephrine release via the activation of pre- and postsynaptic a_2_-adrenoceptors in the locus coeruleus [2,13,14,17,35]. In our study, we observed that the apnea duration was significantly longer in the propofol group than in the thiopental group (296 s vs. 172 s). Since dexmedetomidine is clinically safe for respiration even at supramaximal plasma levels, the difference may be attributed to the fact that the duration of apnea in both groups is largely a result of the depressant effect of propofol or thiopental [7].) Moreover, the longer apnea duration in our propofol group compared to previous studies may be related to the higher dose of propofol used for induction [2,13,17], our use of premedication (midazolam) [2,13,17], the slower injection of dexmedetomidine (10 min vs. 2 min) [2,13], the higher mean patient age [13,17], and the use of different starting points to calculate apnea duration [4,18].

In our study, we found that the induction time (defined as time to loss of the eyelash reflex) was significantly shorter with propofol than thiopental (28.2 s and 37.6 s, respectively), which is consistent with previous reports [8,9] and the results of a study by Afridi et al. [9] in which a similar induction protocol was applied (41.7 s and 51.1 s, respectively). The shorter mean time to lash reflex loss in both groups in our study may be related to the higher proportion of female patients and/or the addition of dexmedetomidine. Although the lack of a control group precluded our analysis, the durations reported for control groups in previous studies suggest that dexmedetomidine administration may have improved the induction time and quality [40,41]. In previous studies, the average LMA placement time was reported as 6 to 38 s [16,27]. The high variability in this time may be related to patient age, practitioner experience, the LMA type, the placement technique, the time calculation method, and the use of induction agents from different groups. However, consistent with previous reports and regardless of the difference in time calculations, we observed that the mean LMA insertion time was significantly shorter with propofol (14.1 s) than thiopental (15.6 s) due to the better insertion conditions [12,16].

LMA insertion can be performed without the use of neuromuscular blocking agents but requires a sufficient depth of anesthesia [29]. Clinical indicators such as the loss of verbal response, loss of motor response to jaw thrust, and apnea are often used to assess the depth of anesthesia in LMA placement [42]. However, although various methods can be used to monitor clinical parameters, such as hemodynamic (HR and blood pressure changes), autonomic (sweating, tears), and somatic (movement or swallowing) indicators, their accuracy is controversial [29,43]. The dose of propofol, dexmedetomidine, and thiopental required to obtain a sufficient depth of anesthesia varies among individuals, so the administration of a fixed dose may lead to airway complications or cardiorespiratory depression associated with shallow or deep anesthesia [14,42,44]. However, during dexmedetomidine infusion, it is difficult to estimate the depth and need for anesthesia using changes mediated by the autonomic nervous system [15]. The BIS is a technique that uses electroencephalography to measure the depth of anesthesia. It has been demonstrated that the BIS enables anesthetic dose titration based on the depth of anesthesia, thereby reducing the incidence of hemodynamic disorders and resulting in improved recovery time [29]. In our study, we monitored anesthesia depth with the BIS and performed LMA placement when the BIS value was below 40 (90 s after induction with additional dosing if necessary). We think this prevented individual differences related to the evaluation of the timing of the LMA insertion, as well as differences in the LMA insertion conditions that can arise with standard doses because of variations in the induction dose needed by different people.

One of the advantages of supraglottic airway devices is that the incidence of pharyngolaryngeal morbidity is lower than with endotracheal intubation (48% vs. 60%) [45,46]. Many variables affect the incidence of LMA-induced pharyngeal morbidity. The most likely cause of airway morbidity with an LMA is trauma during insertion [27]. Sore throat is one of the important determinants of hospital stay, patient dissatisfaction, and postoperative morbidity and is the eighth most common postoperative adverse effect after general anesthesia [22,27,45]. It has been reported that the incidence of LMA-related throat pain is multifactorial and can vary according to the anesthesia depth at the time of insertion, the use of neuromuscular blocking agents, the insertion method, the experience of the practitioner, the number of attempts, the use of a large LMA, high cuff volume or cuff pressure, anesthesia duration, the type of postoperative analgesia, the increase in cuff pressure due to N_2_O diffusion through the cuff wall, and the presence of a heat and moisture exchanger in the circuit [22,28,33,45]. The LMA cuff is inflated with room air, but cuff volume and pressure may increase during anesthesia because the silicone rubber material of the cuff readily absorbs volatile anesthetic agents and nitrogen oxide (N_2_O) [28]. Burgard et al. [33] reported that there was a significant increase in cuff pressure during the first 60 min after an LMA insertion, possibly due to N_2_O diffusion. They stated that postoperative sore throat could be significantly reduced by continuously monitoring cuff pressure for the first three minutes after an LMA insertion and lowering the cuff pressure without serious gas leakage. Chia et al. [22] compared thiopental and propofol and reported that propofol provided lower rates of early pharyngeal morbidity at postoperative 2 h, including sore throat (13% vs. 24%) and dysphagia (3% vs. 15%), due to greater suppression of laryngeal reflexes and better insertion conditions. Venugopal et al. [45] and Rieger et al. [46] compared LMA and endotracheal intubation in terms of the frequency of pharyngolaryngeal morbidity and reported dysphagia after an LMA insertion in 33% and 23.8% of patients, sore throat in 35% and 27%, and blood in 38.3%, respectively. In our study, although there was no significant difference between the propofol and thiopental groups in terms of the incidence of dysphagia (12.5% and 17.9%, respectively), presence of blood (10% and 18.4%, respectively), or sore throat (17.5% and 28.9%, respectively), the tendency toward less pharyngeal morbidity with propofol may be attributable to the better LMA insertion conditions and fewer insertion attempts. The lower incidence of pharyngolaryngeal morbidity in our study compared to previous reports may be related to the fact that LMA insertion was performed at a sufficient anesthesia depth, with a fixed cuff pressure of 60 mmHg, and by an experienced practitioner.

One of the limitations of our study is that our results are only valid for premedicated, ASA I-II, elective surgery patients between the ages of 18 and 65. The hemodynamic response to the combination of 1 μg·kg^−1^ dexmedetomidine with propofol or thiopental may differ in ASA III or IV patients, especially those with severe heart disease, or in unpremedicated or anxious patients such as those undergoing emergency surgeries. However, since the pharmacokinetics and pharmacodynamics of dexmedetomidine, propofol, and thiopental change with age, the doses we used are not applicable to older adults [4,14,42,44]. The absence of a control group can be considered another limitation of our study. However, thiopental and propofol administration alone may result in an increased risk of airway trauma and inadequate ventilation because pharyngolaryngeal reflexes may not be sufficiently suppressed. Therefore, we did not include a control group for ethical reasons.

## 5. Conclusions

The results of our study showed that in healthy, premedicated patients with appropriate airway anatomy, 1 μg·kg^−1^ dexmedetomidine followed by thiopental provided optimal and/or acceptable LMA placement conditions at similar rates to the dexmedetomidine/propofol combination. In addition, although the differences were not clinically significant, thiopental was associated with longer time to eyelash reflex loss and LMA placement time, shorter apnea duration, more stable hemodynamics without an increased risk of hypotension, and a lower rate of optimal LMA insertion conditions compared to propofol. These findings suggest that the administration of dexmedetomidine with thiopental may be an alternative to propofol, especially in outpatients and/or cases where respiratory depression is undesired in terms of airway safety. However, regarding the risk of bradycardia with propofol [2,18,26,31,36], we believe that a lower dose of dexmedetomidine or a slow infusion of the recommended dose, as well as premedication with an anticholinergic agent before induction with propofol if not contraindicated, would be more appropriate in terms of providing a more stable hemodynamics.

## Figures and Tables

**Table 1 medicina-61-00783-t001:** LMA insertion condition scale.

	Insertion Conditions		
Variable	Excellent	Good	Poor
Jaw opening	Complete	Partial	None
Ease of LMA insertion	Easy	Difficult	Impossible
Patient responses			
Swallowing	None	Minor	Severe
Coughing/gagging	None	Minor	Severe
Head and body movement	None	Minor	Severe
Laryngospasm	None	Partial	Complete

LMA insertion conditions: optimal, all responses are excellent; acceptable, all responses are excellent or good; poor, the presence of one or more.

**Table 2 medicina-61-00783-t002:** Demographic characteristics of patients.

			Group P (*n* = 40)			Group T (*n* = 40)			*p*-Value
Age (yr)	Mean ± SD		45.3	±	10.4	43	±	10.3	0.321 ^1^
Sex	Female	*n*-%	18		45	19		47.5	0.823 ^2^
	Male	*n*-%	22		55	21		52.5	
Weight (kg)	Mean ± SD		71.2	±	13.16	70.2	±	10.1	0.690 ^1^

^1^ Independent sample *t* test; ^2^ Pearson chi-square test.

**Table 3 medicina-61-00783-t003:** Characteristics of induction and LMA use duration.

Variables		Group P (*n* = 40)	Group T (*n* = 38)	*p*-Value
LED (s)	Median (Min–Max)	28 (25–34)	38 (30–41)	<0.001 ^1^
AD(s)	Median (Min–Max)	293.5 (230–406)	160 (120–270)	<0.001 ^1^
ID(s)	Median (Min–Max)	14 (11–18)	16 (13–18)	<0.001 ^1^
UD (min)	Median (Min–Max)	51 (31–69)	45 (34–66)	0.072 ^1^

^1^ Mann–Whitney U; LMA insertion duration: ID; apnea duration: AD; LMA use duration: UD; loss of eyelash reflex duration: LED.

**Table 4 medicina-61-00783-t004:** Patient response to laryngeal mask insertion.

	Group P (*n* = 40)		Group T (*n* = 40)		*p*-Value
	*n*	(%)	*n*	(%)	
LMA insertion conditions					0.025 ^1^
Excellent	25	(62.5)	16	40	
Good	15	(37.5)	22	55	
Poor	0	(0)	2	5	
LMA ease of insertion					0.263 ^2^
Easy	38	(95)	34	85	
Difficult	2	(5)	4	10	
Impossible	0	(0)	2	5	
Mouth opening					0.019 ^1^
Complete	31	(77.5)	22	55	
Partial	9	(22.5)	18	45	
None	0	(0)	0	0	
Swallowing					0.330 ^1^
None	36	(90)	33	82.5	
Minor	4	(10)	6	15	
Severe	0	(0)	1	2.5	
Coughing/gagging					1.00 ^2^
None	39	(97.5)	39	97.5	
Minor	1	(2.5)	1	2.5	
Severe	0	(0)	0	0	
Head and limb movement					0.133 ^1^
None	32	(80)	26	65	
Minor	8	(20)	13	32.5	
Severe	0	(0)	1	2.5	
Laryngospasm					0.494 ^2^
None	40	(100)	38	95	
Partial	0	(0)	2	5	
Complete	0	(0)	0	0	
Number of attempts					0.228 ^2^
I	37	(92.5)	32	80	
II	3	(7.5)	6	15	
III	0	(0)	2	5	

^1^ Pearson chi-square; ^2^ Fisher’s exact test.

**Table 5 medicina-61-00783-t005:** Hemodynamic changes after induction.

	Group P (*n* = 40)	Group T (*n* = 38)	*p*-Value
% change from baseline	Median (Min–Max)	Median (Min–Max)	
HR (bpm)			
1 min before	−18.8 (−36.1–(−8.6))	−12.5 (−19.6–(−1.4))	<0.001 ^2^
1 min after	−17.4 (−40.2–(−1.8))	−9.7 (−19.4–3.4)	<0.001 ^2^
2 min after	−16.4 (−43.3–8.8)	−11.5 (−21.4–3.2)	<0.001 ^2^
3 min after	−15.1 (−43.3–8.8)	−14.1 (−22.5–3.2)	0.021 ^2^
4 min after	−14.1 (−44.3–10.5)	−14.9 (−22.5–0)	0.944 ^2^
5 min after	−13.8 (−46.4–12.1)	−16.6 (−23.5–0)	0.379 ^2^
SAP (mmHg)	Median (Min−Max) Mean ± SD	Median (Min−Max) Mean ± SD	
1 min before	−8.4 (−19.6–(−7.3))	−17.9 (−26.7–(−4.6))	<0.001 ^2^
1 min after	−12.2 (−22.8–(−8.8))	−13.1 (−24–(−0.9))	0.664 ^2^
2 min after	−13.8 (−22.2–(−10.9))	−12.7 (−27.3–(−5.9))	0.497 ^2^
3 min after	−18.8 (−27.9–(−14))	−15.3 (−28.7–(−9.1))	<0.001 ^2^
4 min after	−19.1 (−29.1–(−16.2))	−16.3 (−28.7–(−9.1))	<0.001 ^2^
5 min after	−21.5 ± 3.0	−17.5 ± 4.7	<0.001 ^1^
DAP (mmHg)	Median (Min−Max) Mean ± SD	Median (Min−Max) Mean ± SD	
1 min before	−10.3 (−14.3–(−6.8))	−13.5 (−29.5–(−1.3))	0.026 ^2^
1 min after	−13.2 (−22.1–(−6.8))	−3.4 (−21.2–0)	<0.001 ^2^
2 min after	−16.7 (−26–(−8.9))	−7.6 (−23.2–(−4.4))	<0.001 ^2^
3 min after	−19.1 (−34.2–(−10.7))	−12.6 (−27.3–(−8.8))	<0.001 ^2^
4 min after	−20 (−35.4–(−10.7))	−15 (−27.3–(−11.8))	<0.001 ^2^
5 min after	−22.7 ± 5.1	−17.5 ± 3.3	<0.001 ^1^
MAP (mmHg)	Median (Min−Max) Mean ± SD	Median (Min−Max) Mean ± SD	
1 min before	−10.1 (−13.8–(−7.5))	−15.8 (−24.5–(−2.6))	<0.001 ^2^
1 min after	−14 ± 2.7	−8.8 ± 3.2	<0.001 ^1^
2 min after	−16 ± 2.8	−11.4 ± 2.9	<0.001 ^1^
3 min after	−20.4 ± 3.3	−14.9 ± 3.1	<0.001 ^1^
4 min after	−21.2 ± 3.2	−16.4 ± 2.8	<0.001 ^1^
5 min after	−16.6 ± 13.7	−15.2 ± 5.2	<0.001 ^1^

^1^ Independent sample *t* test and ^2^ Mann–Whitney u test.

**Table 6 medicina-61-00783-t006:** The incidence of pharyngolaryngeal morbidity and adverse events in each group.

	Categories	Group P (*n* = 40)		Group T (*n* = 38)		*p*-Value
Presence of blood	1%	36	90	31	81.6	0.285 ^1^
	2%	4	10	5	13.2	
	3%	0	0.0	2	5.3	
Recovery sore throat	0%	33	82.5	27	71.1	0.230 ^1^
	1%	4	10	5	13.2	
	2%	2	5	5	13.2	
	3%	1	2.5	1	2.6	
	4%	0	0.0	0	0.0	
Complications	No	31	77.5	34	89.5	0.156 ^1^
	Bradycardia	6	15 *	0	0.0	
	Hypotension	2	5	4	10.5	
	Hypotension and Bradycardia	1	2.5	0	0.0	
Recovery dysphagia	No	35	87.5	31	82.1	0.469 ^1^
	Yes	5	12.5	7	17.9	

^1^ Pearson chi-square and * comparisons that were performed between the presence and absence of complications.

## Data Availability

The data presented in this study are available on request from the corresponding author due to privacy, legal, or ethical reasons.

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
