# Peer review of "Comparison of the Effects of Propofol–Dexmedetomidine and Thiopental–Dexmedetomidine Combinations on the Success of Classical Laryngeal Mask Airway Insertions, Hemodynamic Responses, and Pharyngolaryngeal Morbidity [Author-notes fn1-medicina-61-00783]"

_medicina, 2025, doi:10.3390/medicina61050783_

Round 1
Reviewer 1 Report
Comments and Suggestions for Authors
A study is presented that evaluated the effect of 1 μg kg-1 dexmedetomidine before induction with 2.5 mg kg-1 propofol or 5 mg kg-1 thiopental under laryngeal mask airway insertion conditions in patients undergoing surgery under general anesthesia lasting less than 2 hours and not requiring intubation. We would appreciate if the authors could respond to the following comments and questions about the study.
- Please designate the study as a clinical trial: title, abstract, and materials and methods.
- Please describe the study design in more detail. Was the design parallel, crossover, factorial, etc.?
- How was the sample size calculated? What formula was used and what were the values?
- It is mentioned that treatments were randomly assigned. Please mention what method was used to perform randomization and if there were any restrictions.
- Was this study single-blind or double-blind? Who was blinded?
- The statistical analysis mentions that the Kolmogorov-Smirnov test was used. Why was this test used to determine the normality of the data and not another test? Does it have any advantage? Perform the analysis of the normality distribution with the Shapiro-Wilks test.
- The flowchart included does not distinguish which groups correspond to each arm.
- The flowchart shows that one treatment performed the statistical analysis with 38 patients. However, the tables included show 40 patients. Why? Specify what were the reasons for the loss of patients.
- Homogenize the tables and the information in general, because they appear moved and even overlapping values are observed.
Author Response
Comments 1: Please designate the study as a clinical trial: title, abstract, and materials and methods.
Response 1: Thank you for pointing this out. We agree with this comment. Therefore, we have updated the manuscript to designate the study as a clinical trial.
Comments 2: Please describe the study design in more detail. Was the design parallel, crossover, factorial, etc.?
Response 2: We agree with this comment. The manuscript has been revised to provide a more detailed description of the study design. The study had a prospective, randomized, double-blind, parallel group design. This design was used to compare the effects of two different drug combinations.
Parallel Design Details:
• The study involved two distinct groups of participants, each receiving a different intervention:
o Group P: Dexmedetomidine + Propofol.
o Group T: Dexmedetomidine + Thiopental.
• This design allowed for the comparison of different treatments over the same time period.
The updated text can be found in the Materials and Methods section
Comments 3: How was the sample size calculated? What formula was used and what were the values?
Response 3: We agree with this comment. The manuscript has been updated to address the lack of a priori sample size calculation. Post hoc power analysis was performed based on the comparison of Group P and Group T at LMA (laryngeal mask airway) insertion conditions. The power of the study was calculated as 99.6% using the online tool available at https://clincalc.com/stats/power.aspx.
Comments 4: It is mentioned that treatments were randomly assigned. Please mention what method was used to perform randomization and if there were any restrictions.
Response 4: The sealed-envelope method was used for random assignment to treatment groups in the study. In this method, information indicating which treatment group (Group P or Group T) each participant would be included in was stored in pre-prepared and sealed envelopes. During randomization, an envelope was selected for each participant, and the participant was assigned to the group specified in the envelope.
Comments 5: Was this study single-blind or double-blind? Who was blinded?
Response 5: This study was conducted as a double-blind trial. The following individuals were blinded to ensure the integrity of the study:
1. Anesthesiologist monitoring the patient: The anesthesiologist responsible for evaluating the patient's responses, inserting the laryngeal mask airway (LMA), and monitoring the study parameters was blinded to the administered drugs.
2. Resident administering the drugs: The drugs were prepared in advance and covered with opaque tape to obscure their contents. The resident injected the drugs behind a drape, ensuring that the anesthesiologist conducting the LMA procedure remained blinded.
3.
Comments 6: The statistical analysis mentions that the Kolmogorov-Smirnov test was used. Why was this test used to determine the normality of the data and not another test? Does it have any advantage? Perform the analysis of the normality distribution with the Shapiro-Wilks test.
Response 6: We agree with this comment and have revised the manuscript to correct the statistical method used for normality testing. The Shapiro-Wilks test was used to assess the normality of the data, and the mention of the Kolmogorov-Smirnov test was a mistake. The updated text clarifies this point and reflects the correct statistical method.
Comments 7: The flowchart included does not distinguish which groups correspond to each arm.
Response 7: We agree with this comment and have revised the flowchart to include clear labels for each group corresponding to their respective arms.
Comments 8: The flowchart shows that one treatment performed the statistical analysis with 38 patients. However, the tables included show 40 patients. Why? Specify what were the reasons for the loss of patients.
Response 8: We agree with this comment and have clarified the discrepancy in the manuscript. During LMA placement, two patients in the thiopental group experienced failed attempts despite two trials. As a result:
• These two patients were included in the pre-placement evaluation tables.
• However, they were excluded from the tables presenting parameters measured after LMA placement.
This explains why Table 2 and Table 4 include data from 40 patients.
Comments 9: Homogenize the tables and the information in general, because they appear moved and even overlapping values are observed.
Response 9: We agree with this comment. The tables and general formatting issues in the manuscript have been thoroughly reviewed and resolved. All tables have been reformatted to ensure uniformity in alignment, spacing, and font size. Overlapping values have been corrected, and the tables now follow a consistent structure throughout the manuscript.
Reviewer 2 Report
Comments and Suggestions for Authors
Pharyngo-laryngeal morbidity needs to be clearly defined
In Methods, I believe that it is necessary to follow an objective criterion for induction of anesthesia, such as decrease in BIS and not loss of eyelash reflex, which is a subjective criterion.
In the Introduction chapter should be reworded:
Row 49-50: "The anesthesiologist's primary responsibility is to ensure that a patent airway is maintained during surgical procedures". The role of the anesthesiologist is not primarily securing the airway, this is only one of many responsibilities.
row 52-53 "oropharyngeal blockage and coverage of the glottic orifice". SGA works on a different principle.
67-68 "its high cost limits its use, especially in developing countries" - the price of propofol has fallen sharply in recent years, and dexmedetomidine is even more expensive than propofol.
70 "is the most commonly used agent in studies using the coinduction method" Needs bibliographic reference
The Methods chapter should specify the period during which the study was performed. The types of surgery and the average duration of surgery should also be stated. It is debatable whether the prolonged awakening with propofol depends on insufficient metabolization or has other causes - this should be stated in the Discussion chapter. Induction anesthetics are unlikely to influence patient awakening.
Sevoflurane concentration should be expressed in MAC, to be corrected.
In Results:
Table 3 the units of measurement should be noted.
In Table 4 percentages should be put in parentheses and it should be clearly stated what each column represents. The same table is modified in the pdf format received by me. To be revised
In Table 5, the change in hemodynamic parameters should be referred to the initial value and expressed in percentages. And these will be compared between the two groups. To completely redo it.
Author Response
Comment 1: Pharyngo-laryngeal morbidity needs to be clearly defined.
Response 1: The definition of pharyngo-laryngeal morbidity has been incorporated into the text, specifying the criteria used for its assessment. These additions have been made in the following sections:
• Rows 29-31: The definition is introduced in the context of the study's objectives.
• Rows 80-84: Detailed criteria for the assessment of pharyngo-laryngeal morbidity are outlined.
Comment 2: In Methods, I believe that it is necessary to follow an objective criterion for induction of anesthesia, such as a decrease in BIS and not the loss of eyelash reflex, which is a subjective criterion.
Response 2:. The time to loss of the eyelash reflex was measured to monitor the induction duration, and the BIS was utilized to determine the suitability for LMA placement.
Comment 3: In the Introduction chapter, the following sentence should be reworded:
Row 49-50: "The anesthesiologist's primary responsibility is to ensure that a patent airway is maintained during surgical procedures." The role of the anesthesiologist is not primarily securing the airway; this is only one of many responsibilities.
Response 3: The sentence has been reworded to reflect the broader scope of the anesthesiologist's responsibilities.
Comment 4: Row 52-53: "Oropharyngeal blockage and coverage of the glottic orifice." SGA works on a different principle.
Response 4: The phrase "oropharyngeal blockage and coverage of the glottic orifice" has been removed to avoid incorrect descriptions of the SGA's mechanism.
Comment 5: Row 67-68: "Its high cost limits its use, especially in developing countries." The price of propofol has fallen sharply in recent years, and dexmedetomidine is even more expensive than propofol.
Response 5: The sentence has been removed to reflect the current pricing trends and avoid misleading information
Comment 6: Row 70: "Is the most commonly used agent in studies using the coinduction method." This statement needs a bibliographic reference.
Response 6: A bibliographic reference has been added to support the statement regarding the common use of the agent in coinduction studies.
Comment 7: The Methods chapter should specify the period during which the study was performed. The types of surgery and the average duration of surgery should also be stated. It is debatable whether the prolonged awakening with propofol depends on insufficient metabolization or has other causes – this should be stated in the text.
Response 7: The Methods chapter has been updated to include the study period and the types of surgery, and the average duration of surgery in the Results chapter.
Comment 8: Induction anesthetics are unlikely to influence patient awakening.
Response 8: We acknowledge that the likelihood of induction anesthetics influencing the patient's recovery time is low. In this context, only the effect of the induction technique on apnea duration was calculated as the time from the loss of spontaneous respiration following the administration of the induction agent to the resumption of spontaneous respiration.
Comment 9: Sevoflurane concentration should be expressed in MAC.
Response 9: The sevoflurane concentration has been corrected and is now expressed in MAC (Minimum Alveolar Concentration) throughout the manuscript to ensure clarity and consistency.
Comment 10: Table 3: The units of measurement should be noted.
Response 10: The units of measurement have been added to Table 3 to ensure clarity and consistency.
Comment 11: In Table 4, percentages should be put in parentheses, and it should be clearly stated what each column represents. The same table is modified in the PDF format received. To be revised.
Response 11: Table 4 has been revised to include percentages in parentheses for clarity.
Comment 12: In Table 5, the change in hemodynamic parameters should be referred to the initial value and expressed in percentages. These changes should also be compared between the two groups. The table needs to be completely redone.
Response 12: Table 5 has been updated to reflect the requested changes.
Round 2
Reviewer 1 Report
Comments and Suggestions for Authors
The authors have taken into account the comments made above, and this version could be accepted.
Author Response
Comments:The authors have taken into account the comments made above, and this version could be accepted
Response: Thank you very much for taking the time to review this manuscript.
Reviewer 2 Report
Comments and Suggestions for Authors
In the Methods chapter, row 94, the interval of the study should be corrected.
Under Results, you wrote that the operations took less than 1 hour, and under Methods, no more than 2 hours. There needs to be a correlation between Methods and Results. Under Results you should note the average duration of the operations.
On all tables, correct Tablo and Maxs with the correct names and abbreviations.
Also for tables, it is not necessary to use mean and median. Instead, the percentage by which the tracked parameters (blood pressure, HR) change at each time point of anesthesia induction should be calculated and these percentages should be compared between the two groups. These alone show the differences between groups.
Author Response
Comparison Of The Effects Of Propofol-Dexmedetomidine And Thiopental-Dexmedetomidine Combinations On The Success Of Classical Laryngeal Mask Airway Insertion, Hemodynamic Response And Pharyngolaryngeal Morbidity
|
Response to Reviewer 2 Comments
|
||
|
1. Summary |
|
|
|
Thank you very much for taking the time to review this manuscript. Please find the detailed responses below and in track changes in the re-submitted files.
|
||
|
|
||
|
Comments 1: In the Methods chapter, row 94, the interval of the study should be corrected. Comments 2: Under Results, you wrote that the operations took less than 1 hour, and under Methods, no more than 2 hours. There needs to be a correlation between Methods and Results. Under Results you should note the average duration of the operations. |
||
|
Response 1 and 2: Thank you for pointing this out.We agree with this comment. In our study, one of the exclusion criteria listed in line 94 of the Methods chapter is operations lasting longer than 2 hours. Therefore, we were unable to amend this statement in line 94. However, in the Results chapter, the statement regarding the types of surgeries and the average duration of the operations for the patients included in the study was corrected in lines 196, 197, and 198 to read: " Patients included in the study underwent operations lasting less than 2 hours (averagely 1 hour) from general surgery, gynecology, obstetrics, or urology."
Comments 3: On all tables, correct Tablo and Maxs with the correct names and abbreviations. |
||
|
Response 3: We agree. Accordingly, we have updated all table names (Tables 1, 2, 3, 4, and 5) and corrected the "Maks" expressions (in Tables 2, 3, and 5) as well as the "Tablo" expression (in Table 5) with the correct names and abbreviations (in Table 3). These changes have been highlighted in red.
|
||
|
Comments 4: Also for tables, it is not necessary to use mean and median. Instead, the percentage by which the tracked parameters (blood pressure, HR) change at each time point of anesthesia induction should be calculated and these percentages should be compared between the two groups. These alone show the differences between groups.
|
|
Response 4: Thank you for pointing this out.We agree with this comment. Table 5 has been revised according to your suggestions
|
